



Atmospheric
Chemistry
and Physics

# Potential new tracers and their mass fraction in the emitted PM$_{10}$ from the burning of household waste in stoves

**András Hoffer[1], Ádám Tóth[2], Beatrix Jancsek-Turóczi[2], Attila Machon[3], Aida Meiramova[2], Attila Nagy[4], Luminita Marmureanu[5], and András Gelencsér[1,2]**

[1]MTA-PE Air Chemistry Research Group, 8200, Veszprém, Hungary
[2]Air Chemistry Research Group, University of Pannonia, 8200, Veszprém, Hungary
[3]Hungarian Meteorological Service, 1181, Budapest, Hungary
[4]Wigner Research Centre for Physics, 1121, Budapest, Hungary
[5]Remote Sensing Department, National Institute of R&D for Optoelectronics, 409 Atomistilor Str.,
Măgurele, Ilfov, Romania

**Correspondence:** András Hoffer (hoffera@almos.uni-pannon.hu)

**Abstract.** The production and use of plastics is increasing rapidly as they are widely used in packaging, construction materials, furniture, foils, etc. As a consequence of their widespread use and often disposable nature, vast streams of plastic waste are continuously generated, a considerable fraction of which are combusted in households worldwide. In this paper, various types of commonly used plastics (PE, PET, PP, PU, PVC, PS, ABS) as well as treated wood samples (LDF, low-density fibreboard) and firewood were combusted separately in a test stove under controlled conditions. The particulates emitted during the combustion test were collected on filters, potential tracers for each waste type were identified by GC-MS, and their relative abundances were determined. The emission factor of 1,3,5-triphenylbenzene was found to be higher for polymers containing aromatic rings in their structure. The application of terphenyls and quaterphenyls as tracer compounds has also been investigated. The trimer of styrene was found to be a potential tracer for the combustion of polystyrene and/or styrene-containing copolymers. Novel tracers were proposed for the burning of PET and furniture plates (LDF), which are among the most widely used waste types burned in households.

# 1 Introduction

Air pollution derived from waste combustion affects human health and may also contribute to climate change (Manisalidis et al., 2020). According to a recent report by Eurostat in Europe, municipal waste was produced at a rate of 483 kg per capita in the year 2016, above the global average of 436 kg per capita (Zaman, 2016). Each year increasing volumes of municipal waste are being generated globally, a considerable fraction of which (1 %–36 %) is of plastic type (Hoornweg and Bhada-Tata, 2012). The amount of domestic waste burned in open fires was first estimated in a modelling study by Wiedinmyer et al. (2014), who found annual global waste production to be 2400 Tg, about 26 % and 15 % of which is being burned openly in households and at dump sites, respectively. The fraction of municipal waste burned in households is difficult to assess, since – being an illegal activity in several countries of the world – it is neither regulated nor controlled and most often goes undetected. Sporadic reports in the media as well as available surveys initiated by NGOs suggest that this practice may be widespread in poverty-stricken regions in many European countries, especially in central–eastern Europe. Emission studies are extremely scarce for burning municipal waste in households; the only comprehensive report on PM$_{10}$ and PAHs emission factors (EFs) for indoor combustion of 12 common types of municipal solid waste has just recently been published (Hof-

fer et al., 2020). This study has revealed that waste burning emits up to 40 times more $PM_{10}$ and 800 times more PAHs than the combustion of dry firewood per mass basis. These findings have highlighted the urgent need for coordinated actions against illegal waste combustion and the extreme health hazard associated with it. Since neither $PM_{10}$ nor PAHs are specific for burning municipal waste in households, the general assessment of the potential contribution of these activities to air pollution in different regions would require the detection of tracers that are highly specific for the burning of different types of municipal waste.

Although polychlorinated dibenzo-*p*-dioxins and dibenzo-*p*-furans (PCDDs/Fs) as well as polychlorinated biphenyls (PCBs) are released from the burning of certain types of municipal waste (Ni et al., 2009; Ferre-Huguet et al. 2006), they may also be emitted by some high-temperature industrial processes (e.g. hot briquetting for non-ferrous recycling materials, sintering of recycled materials; Fiedler, 1996), and – perhaps more importantly – by residential wood burning and road transport (Lavric et al., 2004).

Phthalate esters are used as additives for softening and enhancing flexibility or elasticity of plastic products in, for example, polyvinylchloride resins, polyethylene terephthalate, polyvinyl acetates, and polyurethanes. These compounds can be released upon the combustion of these plastics, since they are not chemically bound to the polymers. However, as phthalate esters are abundant in many plastics, they are not specific; therefore it is difficult to avoid cross-contamination during sampling and analysis, and the high volatility of these compounds also affect particulate-phase measurements. Diphenyl sulfone and dioctyl adipate were also suggested as tracers for waste burning (Giri et al., 2013).

Terephthalic acid (1,4-benzenedicarboxylic acid, TPA) is an important additive in making polyester fibre and PET (polyethylene terephthalate). For the open-air burning of roadside litter and plastic bags Simoneit et al. (2005) found that its emission factor was in the range between 176 and 5033 $\mu g\,g^{-1}$ PM (micrograms per gram of particular matter) [CE1]. In a highly polluted city in India the mass concentrations of TPA were found to be in the range of 7.6–168 $ng\,m^{-3}$, being the most abundant diacid near an open waste burning site (polyethylene, foam, paper, packaging material, and clothes) (Kumar et al., 2015). Phthalic acid was also found in waste burning smoke in which it may be produced by photo-degradation of di-(2-ethylhexyl) phthalate (DEHP) and terephthalic acid (Kumar et al., 2015). However, phthalic acid was found in high abundance in the smoke from burning low-quality coal in China (He et al., 2018), and it was also found to be emitted from other (traffic-related or biomass burning) sources (Fraser et al., 2003; Fine et al., 2004; Al-Naiema and Stone, 2017), which challenges its use as a specific tracer.

Nitroarenes such as 1-nitropyrene, 1,3-dinitropyrene, 1,6-dinitropyrene, and 1,8-dinitropyrene were also found in particulates from the burning of different types of plastics (PVC, PET, PS, PE) in concentrations of 0.2–76.8 $ng\,g^{-1}$ PM (Lee et al., 1995). However, they were also detected in the exhaust of diesel engines (Keyte et al., 2016).

Fu and Kawamura (2010) suggested that the open burning of plastics may be a significant source of bisphenol A in urban regions based on the significant correlation with the concentrations of other specific waste burning tracers, 1,3,5-triphenylbenzene (135-TPB) and tris(2,4-di-tert-butyl-phenyl) phosphate (TBPP).

135-TPB was proposed as a unique tracer for garbage burning emission (Simoneit et al., 2005). The formation of 135-TPB in smoke from the burning of plastics is linked to the self-condensation reaction of acetophenone which emerges from the styrene chain terminator added to formulations during ethylene polymerization (Simoneit, 2015). Several authors determined the concentration of 135-TPB in particulate samples (Simoneit et al., 2015, and references therein). The atmospheric concentration of 135-TPB was found to be elevated (by a factor of 3–10) near a plant of electronic waste recycling applying high-temperature operations (Gu et al., 2010).

Organophosphate esters (OPEs) are produced in large quantities and used as flame retardants, plasticisers, and anti-foaming agents in products such as furniture, textiles, cables, building and insulation materials, paints, floor polishes, and electronic equipment. Thus, combustion of mixed household waste releases OPEs with the particulates (Liu and Mabury, 2018). Organophosphates are not found in smoke from other emission sources such as coal combustion, biomass burning, or vehicular emission, so they may be considered to be specific tracers for thermoplastic combustion. Tris(2,4-di-tert-butylphenyl) phosphate (an oxidation product of the antioxidant Irgafos 168) has been reported to occur at high concentrations in particulates from plastic combustion (Simoneit et al., 2005). It should be noted that Irgafos 168 and its oxidation product (TBPP) were also identified in the laboratory workplace, which prevented the study of its leaching from plastics (Hermabessiere et al., 2020).

Our knowledge on the specific tracer compounds are still quite limited, since in spite of the fact that many compounds may be emitted during waste burning, most of them are not unique tracers for the process. Among them 135-TPB is used most commonly as a specific tracer for (mixed) waste burning, and even its emission factor was determined for the open burning of mixed waste (Simoneit et al., 2005), and more recently for the open burning of mixed waste, foil wrappers and plastics (Jayarathne et al., 2018) as well as for the co-burning of PET and PE with firewood in a residential boiler (Tomsej et al., 2018).

In our study, we combusted different types of plastic waste separately in a test stove under controlled conditions and collected the particulates emitted on filters. The samples were analysed by GC-MS for 135-TPB as well as for other known tracers, and their mass fraction in the emitted $PM_{10}$ (their relative emission factors) were also determined. The filter

samples were also screened for potentially new and specific tracers with relative mass fractions that may help assess the contribution of waste burning to particulate ($PM_{10}$) pollution in the atmosphere.

## 2 Experimental

### 2.1 Filter sampling

The details for the laboratory waste burning experiment can be found in Hoffer et al. (2020). Briefly, 12 different waste types – 7 plastic types: acrylonitrile butadiene styrene (ABS), polyethylene (PE), polyethylene terephthalate (PET), polypropylene (PP), polystyrene (PS), polyurethane (PU), polyvinyl chloride (PVC); 2 types of treated wood: furniture panel made from low-density fibreboard (LDF) and oriented strand board (OSB); and other waste types such as tire, paper (PAP), and different rags (RAG) – as well as firewood was burned in a stove used for residential heating. The stove used was a Servant S114 cast iron stove with a heating capacity of 5 kW. It is a commercially available, low-cost heating appliance, with a simple stove design; thus it is readily available for the general public for heating their homes with firewood. The simple construction design of the stove also allows the users to combust different items of solid waste in it. This type of stove is widely used in central–eastern Europe, and possibly elsewhere in Europe. Thus our experiments may be considered representative for the burning of small amounts of household waste in low-capacity stoves used for domestic heating in Europe. A fraction of the produced $PM_{10}$ particles (after dilution with ambient air) was collected on quartz filters (Advantec QR-100 $\varnothing$ 150 mm) by a high-volume sampler (Kalman Systems). The filters were weighted before and after the sampling to determine the collected $PM_{10}$ mass gravimetrically. These measurements were performed according to the European standard (MSZ EN 12341: 2014). To address the differences of the stoves and/or the burning parameters used by the population, the burning experiments were performed by setting different air supply ratios at the stove. All samples (except the paper samples) were burned using the low and high air intake and also by the combination of these two air intake settings. Thus at least three different samples were produced for a given waste type, except for the paper samples which were burned only in the combined air supply mode.

For each waste type, one blank sample was also collected. During the blank measurements only charcoal was burning. The sampling times of the blanks were comparable to those of the samples. The limit of quantification (expressed in nanograms of compound per filter) was defined as the sum of the average of the blanks and 10 times the standard deviation of the blanks.

All the data reported here refer to the particular experimental set-up as described in detail in Hoffer et al. (2020),

i.e. "average" burning conditions in terms of air supply and combustion temperature, dilution ratio, etc. Measured concentrations of organic combustion tracers in particulates are highly sensitive to experimental conditions because they are formed in pyrolysis processes, and they are affected by condensation and volatilization, and to some extent by thermochemical transformations. To minimize memory effects between the combustion tests, the stove and sampling line were heated up to about 700 °C (measured at about 50 cm above the outlet of the stove).

The relative EF ($\mu g\,g^{-1}$ PM) of the given compound was calculated from the amount of the compound (or that of its surrogate compound; see below) determined by GC-MS and from the mass of $PM_{10}$ measured gravimetrically. The absolute EF ($mg\,kg^{-1}$) for a given compound was calculated by taking into account the dilution factor in the sampling line and the mass of the waste burned (Hoffer et al., 2020). In the results section, the relative EF of the different components are discussed, but the absolute EFs are also given in the Supplement (Table S2).

Ambient samples were collected in winter on 14, 17, and 25 February 2020 in Magurele (Atmosferei Street), Romania, located about 12 km from the Bucharest centre, in a suburban zone, surrounded by agricultural and residential areas. The $PM_{10}$ samples were collected on quartz filters (Advantec QR-100) with DIGITEL DHA-80 high-volume samplers. The sampling time was 24 h, starting from midnight for each sample. The meteorological situation was characterized by a high-pressure regime on 14 and 17 February, and with low pressure on 25 February. The temperature values were above 0 °C during the daytime and were slightly below 0 °C during the night.

### 2.2 GC-MS analysis

The concentration of organic tracer compounds in the collected aerosol samples was determined by GC-MS as follows. Known area of the exposed filter (452 or 904 $mm^2$) was spiked with 150 µL recovery standard solution (*p*-terphenyl-d14, Sigma-Aldrich) and dried under a nitrogen stream. The samples were then extracted in three steps with $2 \times 4$ mL and $1 \times 3$ mL dichloromethane – methanol mixture ($2:1\ v/v$) using a Vortex agitator ($3 \times 15$ min, $750\ min^{-1}$). The extracts were filtered by syringe filters (Millex-LCR, hydrophilic PTFE membrane, 0.45 µm pore size, 13 mm diameter, Millipore), and they were dried under gentle stream of nitrogen in vials with a volume of 2 mL. The average recovery was found to be 93 % (RSD = 8 %).

Before the GC-MS measurement of non-polar tracer components, the extracts were re-dissolved by adding a 100 µL internal injection standard (ISTD) solution (chrysen-d12, 98 atom % *D*, Sigma-Aldrich) and 150 µL dichloromethane-methanol mixture ($2:1\ v/v$). About 1 µL of the samples was injected manually into the injector (temperature: 300 °C, splitless mode) of an Agilent 6890N gas

chromatograph coupled to an Agilent 5973N mass spectrometer. The separation was performed on a capillary column (DB-5MS UI, 30 m × 0.25 mm × 0.25 μm, Agilent). The temperature program was adopted from Simoneit et al. (2005). The concentrations of the components were determined in SIM mode. After the measurements, a recovery standard (sedoheptulose anhydride monohydrate, Sigma-Aldrich) was added to selected samples, and after drying they were derivatized with 100 μL of BSTFA-TMCS (N,O-bis(trimethylsilyl)trifluoroacetamide–trimethylchlorosilane, 99:1, Sigma-Aldrich) and 100 μL of pyridine (anhydrous, Merck) at 80 °C for 1 h. About 1 μL of the derivatized samples was injected manually into the injector (temperature: 280 °C, splitless mode) of GC-MS. The separation was performed on the same capillary column used for the non-polar components. During the measurements, the column was heated at 60 °C for 1 min; then it was heated up to 300 °C with a temperature rate of 10 °C min$^{-1}$ and maintained for a further 5 min. The measurement lasted for 30 min. For the quantitative analysis SIM mode was used. The calibration of the instrument was performed using available standards (*p*-terphenyl, 135-TPB, melamine, 2,4,6-triphenyl-1-hexen (SSS), terephthalic acid). Based on the recoveries and repeated measurements the estimated uncertainty of the analytical measurements is about 20 %. The amounts of the o- and *m*-terphenyls were expressed in the amount of *p*-terphenyl. The amount of the 1,2,4-triphenyl benzene and the quaterphenyls as well as the 2-(benzoyloxy)ethyl vinyl terephthalate (2-BEVT) were expressed in 135-TPB, and the amount of the ABS pyrolysis compounds (2-methylene-4-phenylheptanedinitrile (ASA), the 2-methylene-4,6-diphenylhexanenitrile (ASS), the 4,6-diphenylhept-6-enenitrile (SSA), and the 2-phenethyl-4-phenylpent-4-enenitrile (SAS)) were expressed in SSS.

## 3 Results and discussions

### 3.1 Potential new tracers for waste burning and their mass fraction in the emitted PM$_{10}$

#### 3.1.1 Terphenyls

The three isomers of the terphenyls (ortho-, meta-, para-) can be easily identified by their intense molecular ion ($m/z = 230$) and by the doubly charged ion at $m/z = 115$. The obtained retention indices of these compounds compared with those in the literature are given in Table S1. Terphenyls were identified in particulates from the burning of several types of plastic waste, but they were found in higher relative concentrations from the burning plastics inherently containing aromatic structures (PET, PS, ABS) (see Fig. 1).

The samples of textiles (RAG) also contained polyesther (PET) besides cotton and polyamide, accounting for the high relative emission factors (mass fraction) of terphenyls during

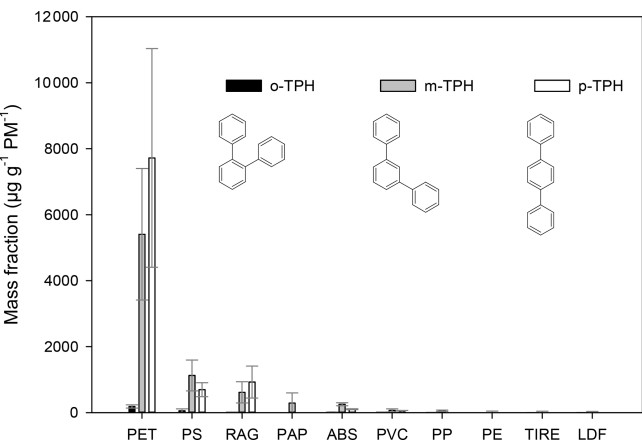

**Figure 1.** The mass fraction of terphenyls (in units of μg g$^{-1}$ PM$^{-1}$) in particulates from the burning of different waste types. The error bars represent the standard deviation of the EF data obtained for a given type of waste under different burning conditions.

their combustion (14, 610, and 920 μg g$^{-1}$ PM for the ortho-, meta-, and para-terphenyl, respectively). The mass fractions of the terphenyls were found to be the highest for the combustion of PET bottles (190, 5400, and 7700 μg g$^{-1}$ PM for the ortho-, meta-, and para-isomer, respectively), and the combustion of PS and ABS yields somewhat lower emission factors (18–56, 250–1100, 100–690 μg g$^{-1}$ PM for the ortho-, meta-, and para-isomer, respectively). Terphenyls have not been reported to be present in particulates released from wood burning (with the exception of liquid wood smoke, Guillen et al., 2000) but have been positively identified in particulates from one lignite combustion sample (Fabbri et al., 2009). Here we also note that the position of the aromatic ring in the polymer structure might determine the structure of the terphenyls formed during the burning (this can also be observed for the quaterphenyls; see below). Our results clearly showed that the *p*-terphenyl is emitted in the largest amounts during PET burning. The aromatic ring in the PET polymer chain is bonded in the para-position (1–4 position), whereas in the other polymers (ABS, PS) it is bonded by a single C–C bond to the polymer chain. Since the ortho-isomer is emitted in low amount during the burning, the ratio of the meta- and the para-isomer might be used to imply the major type of plastics burned. In the case of the PET and RAG burning, the concentration ratio of the emitted *p*-terphenyl to *m*-terphenyl varied between 1.1 and 1.7 (the average was 1.5), whereas this ratio was 0.5–0.8 (on average 0.6) and 0.4 for PS and ABS, respectively. For the other waste types only the meta-isomer was identified, and the concentration of the *p*-terphenyl was below the quantification limit (resulting in a lower ratio of *p*-terphenyl to *m*-terphenyl in ambient samples affected by the burning of PS and/or ABS). Since the proposed aromatic compounds are not unique tracers for the burning of a given waste type, thus the application of their relative ratios in ambient particulate samples for emission

source apportionment would require highly complex statistical calculations in combination with the use of highly specific tracers.

Since the contribution of other sources to the concentrations of terphenyls in atmospheric particulates is low, they may be used as atmospheric tracers for the combustion of plastic waste, in particular PET, PS, and ABS. The mass fractions of terphenyls in the emitted $PM_{10}$ for wood and wood-based waste burning (OSB and LDF) were below the limit of quantification.

### 3.1.2  2,4,6-Triphenyl-1-hexen (styrene trimer, SSS)

The 2,4,6-triphenyl-1-hexen (styrene trimer, SSS) has been positively identified and quantified in the filter samples by the GC-MS analysis of authentic standard (FUJIFILM Wako Chemicals). The presence of this compound was verified by the co-eluting fragment ions of 91, 117, 194, and 207. Tsuge et al. (2011) identified SSS by direct pyrolysis gas chromatography mass spectrometry (Py-GC-MS) of PS and ABS, as well as in other styrene-containing materials such as a styrene–methyl acrylate copolymer, styrene–maleic anhydride copolymer, and styrene–divinylbenzene copolymer. In our experiments, the styrene trimer was found primarily in particulates from the combustion of polystyrene (Fig. 2). Its mass fraction showed variations between 56 and $1400\,\mu g\,g^{-1}$ PM, depending on the conditions of combustion, the average being $2900\,\mu g\,g^{-1}$ PM. SSS was also found in particulates from the combustion of LDF with relative EFs ranging from < LOQ to $1400\,\mu g\,g^{-1}$ PM (on average $380\,\mu g\,g^{-1}\,PM^{-1}$). For paper burning, the EFs were 97 and $670\,\mu g\,g^{-1}$ PM, and waxy, coated papers yielded the higher relative EFs for this compound. The direct Py-GC-MS study (Tsuge et al., 2011) on ABS found SSS only in small amounts, similarly to our study in which combustion of ABS samples yielded relative EF of SSS below the limit of quantification.

It should be noted that traces of SSS have been detected in nearly all of the filter samples as well as in the blanks, implying that it might be a ubiquitous contaminant, so precautions must be taken during its determination and quantification.

Taking into account the fact that a typical "composite" waste mix that is illegally burned in households in Europe includes a large fraction of waste paper and LDFs due to their mass availability, high density, and high calorific values, as well as PS due to its high volume and difficulties in its recycling and disposing, we suggest that the presence of SSS in atmospheric particulates may be a ubiquitous tracer for waste burning in Europe.

### 3.1.3  2-(Benzoyloxy)ethyl vinyl terephthalate (2-BEVT)

Pyrolysis of the PET polymer results in the formation of various compounds, of which 2-(benzoyloxy)ethyl vinyl tereph-

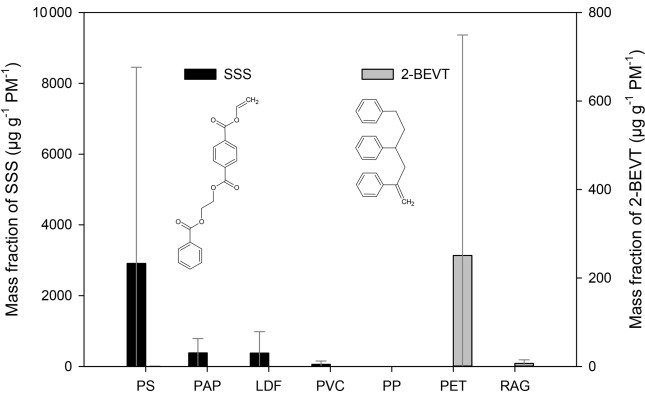

**Figure 2.** The mass fraction (in units of $\mu g\,g^{-1}$ PM) of styrene trimer (SSS) and that of the 2-(benzoyloxy)ethyl vinyl terephthalate (2-BEVT) from the burning of different waste types. The error bars represent the standard deviation of the EF data obtained for a given type of waste under different burning conditions.

thalate was identified in most PET-containing samples (PET and RAG) (see Fig. 2). The identification of this compound was based on the comparison of the mass spectra and the retention index reported in the literature (Tsuge et al., 2011) with the ones obtained during the measurements. The emission factor of this compound was exceptionally high ($1000\,\mu g\,g^{-1}$ PM) from one PET burning experiment during which the average temperature of the flue gas (measured 50 cm above the outlet of the stove when the burning of the waste pieces was started) was the lowest ($\sim 185\,°C$). On the other hand, this compound was not detected in PET samples that were collected using a high air supply ratio and high flue gas temperature (around $280\,°C$). This clearly indicates that this compound forms during pyrolysis, and thus its EF highly depends on the burning conditions. In another PET sample when the air supply was set lower, but the temperature was higher, the EF of 2-BEVT was 2 orders of magnitude lower, $4.2\,\mu g\,g^{-1}$ PM. The average relative EF of 2-BEVT for PET bottles was found to be $250\,\mu g\,g^{-1}$ PM. Interestingly this compound was identified in all three RAG samples in which the EF was also closely related to the settings of the air supply and/or the temperature of the burning. The EF of this compound in the RAG samples varied between 0.36 and $16\,\mu g\,g^{-1}$ PM (on average $6.9\,\mu g\,g^{-1}$ PM).

### 3.1.4  Melamine

Melamine ($C_3H_6N_6$, heterocyclic aromatic compound) is the component of the melamine-formaldehyde resin used mainly in the furniture industry for the surface coating of fibreboard plates, but laminate flooring and some other objects (e.g. kitchen tools) might also contain this polymer. The formation of melamine upon the pyrolysis of melamine-formaldehyde resins was proven by a direct Py-GC-MS study (Tsuge et al., 2011). Melamine was found almost exclusively in par-

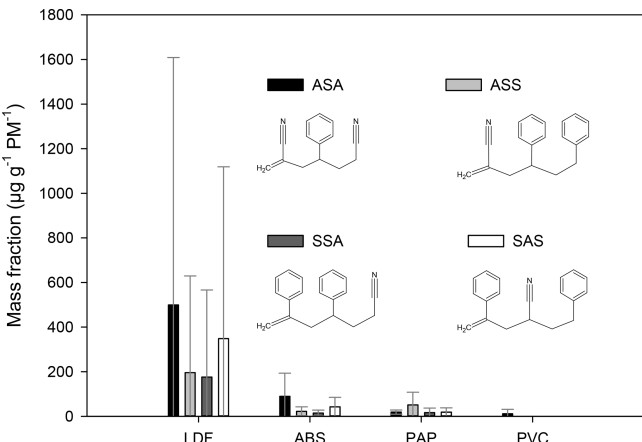

**Figure 3.** The mass fraction of the compounds (in units of $\mu g\,g^{-1}$ PM) emitted during the burning of styrene-containing copolymers. The error bars represent the standard deviation of the EF data obtained for a given type of waste under different burning conditions.

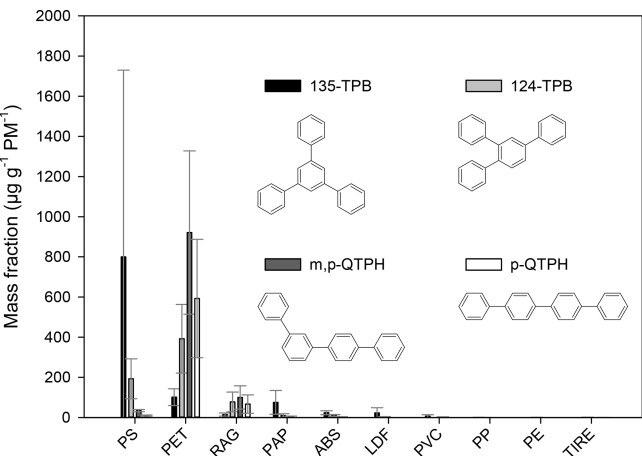

**Figure 4.** The mass fraction (in units of $\mu g\,g^{-1}\,PM^{-1}$) of triphenylbenzenes and quaterphenyls from the burning of different waste types. The error bars represent the standard deviation of the EF data obtained for a given type of waste under different burning conditions.

ticulates from the combustion of LDF samples, with relative EFs in the range between ∼ 50 and ∼ 41 000 $\mu g\,g^{-1}$ PM (on average ∼ 19 000 $\mu g\,g^{-1}$ PM). In all other investigated samples, the levels of melamine were around that of the blanks (and below the detection limit). Although the long-term atmospheric stability of melamine is likely quite low since its molecule contains three reactive amine groups, if detected, it may still be a useful particulate tracer for local waste burning in wintertime. However, its quantitative application may be strongly limited due to its atmospheric instability and the wide variation in the melamine-formaldehyde resin contents of commercially available LDFs.

### 3.1.5 Specific tracers for burning of styrene copolymers

One of the styrene copolymers is the ABS (acrylonitrile butadiene styrene), which is prepared by the copolymerization of acrylonitrile, butadiene, and styrene. The ABS is used in automotive industries, but the housing of the electronic equipment and the edge foil of the furniture plates (LDF) are also frequently made from this material. In the direct pyrolysis of samples containing ABS, the following hybrid trimer compounds consisting of styrene and acrylonitrile units have been identified based on the mass spectra and the retention index reported by Tsuge et al. (2011) (see Table S1). The ASA, ASS, SSA, and SAS have been identified in the samples of ABS, LDF, and PAP (Fig. 3).

Among the samples collected during the ABS burning the EF of the ASA was the smallest in the sample collected with high air intake (its amount was below the LOQ), but on average the EF of ASA was the largest in the case of ABS burning (Fig. 3). In the LDF samples the concentration of the ABS pyrolysis products was highly variable. They were below the LOQ in one sample (collected with a low air in-

take setting), whereas in another LDF sample their relative EFs were among the largest, varying between about 870 and 2500 $\mu g\,g^{-1}$ PM. The large variation of the EF of ABS pyrolysis product in the LDF samples is clearly caused by the inhomogeneity of the composition and ABS content of the burned materials. Similarly to the ABS in the case of the LDF burning, the relative EF of the ASA was the largest followed by the SAS (Fig. 3). In the samples from paper burning the EF of the ABS pyrolysis products was relatively low: for the individual components it varied between about 3 and 91 $\mu g\,g^{-1}$ PM. In the paper industry, sizing is an important process during which the surface properties of the papers are altered. Sizing agents can be applied either by blending or on the surface. Sizing agents (polymers) often contain styrene; for example the styrene–butyl acrylate copolymer is the most common surface sizing agent (Hagiopol and Johnston, 2012). On the other hand, the binders used in paper industry (to bind the coating pigment together and these to the base paper surface) are emulsion of polymers which are the copolymers of several monomers: styrene, butadiene, acrylic esters, vinyl acetate, and acrylonitrile (Bajpai, 2015). This might explain why these compounds have been found in the paper samples.

### 3.2 Previously identified tracer components and their isomers

Triphenylbenzenes and quaterphenyls are composed of four aromatic rings attached to each other by C–C bonds forming non-condensed structures. Their presence in the chromatograms can be identified by the intense molecular ion at $m/z = 306$. In this study, the identification of quaterphenyls was based on comparison of their mass spectra and the calculated retention indices with literature data (see Table S1). The presence of the 135-TPB, 124-TPB, and *p*-QTPH was iden-

tified by authentic standards. The mass fractions of 1,3,5-triphenylbenzene were determined for the combustion of all waste types (Fig. 4). The highest emission factors were found for the materials containing aromatic rings in their polymeric chain (800 and 100 $\mu g\,g^{-1}$ PM for PS and PET, respectively), but for the PAP, LDF, and ABS burning its emission is also not negligible (on average 23–75 $\mu g\,g^{-1}$ PM). Jayarathne et al. (2018) found similar results (12–51 $\mu g\,g^{-1}$ PM) for the open burning of mixed waste (containing cardboard, chip bags, food waste, paper, plastic bags, clothes, diapers, and rubber shoes). Tomsej et al. (2018) have studied the emission factor of 135-TPB from the co-combustion of PE plastics (shopping bags) and PET bottles with wood (beech logs) in a 20 kW boiler. The mass fraction of 135-TPB for the mixed PET burning was 13 $\mu g\,g^{-1}$ PM, whereas for the PE burning it was 3 $\mu g\,g^{-1}$ PM.

The quaterphenyls were emitted in the largest amounts during the burning of PET-containing waste. Their emission factor varied between 33 and 1200 $\mu g\,g^{-1}$ PM for individual samples, the average relative EF for the $m, p$-quaterphenyl being 920 and 100 $\mu g\,g^{-1}$ PM, and that of the $p$-quaterphenyl 590 and 67 $\mu g\,g^{-1}$ PM for the PET and RAG, respectively.

Similarly to the terphenyls (which contain three non-condensed aromatic rings) the ratio of the quaterphenyls and triphenylbenzenes (containing four non-condensed aromatic rings in linear and in branched structure, respectively) can also be used as an indication for the burning of different plastic types.

The average ratio of the $m, p$-quaterphenyl ($m, p$-QTPH) to 124-TPB was lower than 1 in the case of the ABS, PS, and PAP (0.3, 0.2, and 0.2, respectively). In the case of the samples collected during the burning of LDF (which also showed the presence of the styrene-containing materials) only the 124-TPB was measured in higher amounts than the LOQ in three out of five samples. In the samples collected during the PET and RAG burning, the average ratios of these two components were 2.3 and 1.3, respectively.

### 3.3 Estimation of the gas-to-particle partitioning of the newly suggested waste burning tracers

The gas-to-particle partitioning of the compounds were estimated using US EPA (2012). The estimated vapour pressures of the identified compounds are in the range between $3.87 \times 10^{-11}$ mmHg at 25 °C and $1.64 \times 10^{-5}$ mmHg at 25 °C; the latter is lower than that for the syringic aldehyde, for example ($6.49 \times 10^{-5}$ mmHg at 25 °C), suggested as a biomass burning tracer by Simoneit et al. (2002), and is comparable with that of terephthalic acid ($1.19 \times 10^{-5}$ mmHg at 25 °C). We also note here that the estimated vapour pressures of SSS and 2-BEVT ($3.34 \times 10^{-7}$ mmHg at 25 °C and $7.51 \times 10^{-7}$ mmHg at 25 °C, respectively) are very close to that of levoglucosan ($3.47 \times 10^{-7}$ mmHg at 25 °C), the most widely used biomass burning tracer compound.

### 3.4 Identification of waste burning tracers in ambient PM$_{10}$ samples

In order to verify the presence of the above tracer compounds in atmospheric particulates, PM$_{10}$ samples collected in Bucharest (Romania) were also analysed. In the region of Bucharest illegal waste burning is an existing environmental problem, according to recent press reports (Reuters, 2021; Romania Insider, 2020). The concentrations of the tracer compounds as well as the intensity ratios of the target and qualifier ions measured in the laboratory samples as well as in the ambient samples are shown in Table 1.

It can be seen that the intensity ratios of the target ion and the qualifier ion of a few tracer compounds (marked in italics in Table 1 for the $m$-terphenyl ($m$-TPH), SSA, and the ASA in one sample) measured in the ambient samples differ considerably from those obtained in the laboratory experiments, but for the majority of the tracer compounds these ion ratios agree well, indicating that co-eluting components do not affect their detection. Here we note that in the mass spectra of compounds dominated by aromatic rings in their structure (e.g. terphenyls, quaterphenyls) the number and the intensity of the fragment ions are low, and thus other components present in the complex matrix of environmental samples easily mask the ratio of the target ion to the qualifier ion.

The fact that all suggested tracers were positively identified in atmospheric PM$_{10}$ samples indicates that in spite of uncertainties regarding their atmospheric stability they may be used for verifying the occurrence of waste burning in the region and providing a lower bound estimate for their contribution to PM$_{10}$ concentrations, at least in winter.

### 4 Summary

Although burning of waste in households is strictly prohibited in all countries in Europe, there is ample evidence that many people regularly burn waste primarily for heating their homes. Since upon the burning of solid waste in stoves exceptionally high amounts of PM$_{10}$ and polycyclic aromatic hydrocarbons (PAHs) are emitted (Hoffer et al., 2020), the assessment of the contribution of illegal waste burning to ambient PM$_{10}$ concentration levels is vitally important. In this work, different types of plastics which are abundant in household waste were burned under controlled conditions in the laboratory with a view to identifying potentially specific tracer compounds and determining their emission factors. It was found that the mass fraction in the emitted PM$_{10}$ (the relative emission factor) of the universal tracer compound (1,3,5-triphenylbenzene) previously suggested for waste burning (Simoneit et al., 2005) strongly depends on the type of the waste burned, being higher for plastics which contain aromatic ring in their structure. Besides 1,3,5-triphenylbenzene its structural isomers (124-TPB

and quaterphenyls) were also identified in the samples. Their presence in ambient PM$_{10}$ may also support the occurrence of waste burning activities. It was shown that the concentration ratio of triphenylbenzenes to quaterphenyls as well as quaterphenyls to terphenyls (especially $m$-terphenyl and $p$-terphenyl) might carry information on the type of plastics burned, for example upon burning of PET larger amounts of the linear quaterphenyls are emitted. New and highly specific tracers for the burning of certain types of plastics have been identified in the laboratory experiments. These compounds were previously reported as direct pyrolysis products of plastics but have never been suggested for use as potential tracers in atmospheric studies. These novel tracers include the trimer of styrene (SSS) which was emitted from the burning of polystyrene, but also from the burning of low-density fibreboard and coated paper (newspaper). Another new suggested tracer is melamine that is released upon the burning of low-density fibreboard, as melamine-formaldehyde resin is often used for the production of this material. 2-(Benzoyloxy)ethyl vinyl terephthalate (2-BEVT) was identified only in particulates from the burning of PET and RAG, with highly variable emission factors. Although other criteria for potential tracer compounds have not been investigated, the fact that all the suggested compounds were also found in ambient particulate samples implies that they may be worth being considered as potential atmospheric tracers, at least indicatively. The most important criteria used in this work were the specificity and the confirmed presence of these compounds in ambient particulate samples. As a matter of course, further studies on their atmospheric stability and gas-to-particle partitioning are required to verify their applicability as waste burning tracers.

*Data availability.* Data used in this study are available from the first author upon request (hoffera@almos.uni-pannon.hu).

*Supplement.* The supplement related to this article is available online at: https://doi.org/10.5194/acp-21-1-2021-supplement.

*Author contributions.* AH, AT, and BJT collected the aerosol samples. AH, AT, BJT, AiM, and AtM performed and/or coordinated the analytical measurements and data evaluation. All authors were involved in the scientific interpretation and discussion of the results as well as in manuscript preparation. All co-authors commented on the paper.

*Competing interests.* The contact author has declared that neither they nor their co-authors have any competing interests.

**Table 1.** Ambient concentrations and ion intensity ratios of different tracer compounds for waste burning identified and quantified in ambient PM$_{10}$ samples collected in Bucharest. Italic fonts indicate $m/z$ ratios that differ significantly from those of the standards.

| | Ambient concentration (ng m$^{-3}$) | | | Target ion/ Qualifier ion | Intensity ratio of target and qualifier ions obtained for the stove samples (and for standards) | Ion intensity ratio | | |
| --- | --- | --- | --- | --- | --- | --- | --- | --- |
| | BUC/M-200214 | BUC/M-200217 | BUC/M-200225 | | | BUC/M-200214 | BUC/M-200217 | BUC/M-200225 |
| $m$-TPH | 0.13 | 0.15 | 0.12 | 230/115 | 13.7 | 7.2 | 7.2 | 11.0 |
| $p$-TPH | 0.14 | 0.17 | 0.11 | 230/115 | 10.4 (10.9) | 10.2 | 8.3 | 9.5 |
| 135-TPB | 2.7 | 3.3 | 3.6 | 306/289 | 8.9 (9.4) | 9.2 | 9.3 | 9.1 |
| 124-TPB | 0.46 | 0.45 | 0.45 | 306/289 | 4.2 (4.1) | 4.3 | 4.2 | 4.3 |
| $m,p$-QTPH | 0.080 | 0.11 | 0.091 | 306/289 | 14.6 | 12.3 | 16.2 | 15.0 |
| $p$-QTPH | 0.046 | 0.071 | 0.059 | 306/289 | 17.6 (16.5) | 15.3 | 14.9 | 16.1 |
| SSS | 5.3 | 5.5 | 4.8 | 91/117 | 3.1 (3.3) | 3.3 | 3.1 | 3.2 |
| 2-BEVT | 0.16 | 0.10 | 0.17 | 297/149 | 3.2 | 3.2 | 3.2 | 3.1 |
| ASA | 3.6 | 3.9 | 4.8 | 144/91 | 3.4 | 3.5 | 3.5 | 3.5 |
| ASS | 1.9 | 0.96 | 2.6 | 117/91 | 0.4 | 2.3 | 3.2 | 3.5 |
| SSA | 0.39 | 0.47 | 1.3 | 144/91 | 2.8 | 0.3 | 0.4 | 0.3 |
| SAS | 1.5 | 0.99 | 2.6 | 170/91 | 1.1 | *1.2* | *1.9* | 2.2 |
| Melamine-TMS | 37 | 37 | 32 | 327/342 | 1.9 | 0.8 | 0.9 | 1.0 |
| PM$_{10}$ (µg m$^{-3}$) | 61.3 | 51.5 | 47.2 | | | *1.8* | 1.9 | 2.0 |

*Acknowledgements.* The authors are thankful for the support of the János Bolyai Research Scholarship of the Hungarian Academy of Sciences.

*Financial support.* This research has been supported by the project "Analysing the effect of residential solid waste burning on ambient air quality in central and eastern Europe and potential mitigation measures" (grant no. 07.027737/2018/788206/SER/ENV.C.3) and the National Research, Development and Innovation Office, NKFIH-471-3/2021, National Multidisciplinary Laboratory for Climate Change.

*Review statement.* This paper was edited by Ivan Kourtchev and reviewed by two anonymous referees.

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

**Remarks from the language copy-editor**

CE1    Please check; the unit has been defined in full. The unit has been edited as the superscripted "–1" alongside PM is not necessary (as it only relates to gram).