# Peer review of "Potential new tracers and their mass fraction in the emitted PM10 from the burning of household waste in stoves"

_Atmospheric Chemistry and Physics, 2021_

## Author Response (AR1)

**Response to Interactive comment of Anonymous Referee #1**

*Comments and questions of the reviewer are in italics*

Authors' answers are in regular typeface

Parts of the answers highlighted in yellow are inserted into the revised manuscript.

The authors thank the referee for the positive review and the comment.

*General comments:*

*This manuscript presents potentially important new marker compounds for the illegal combustion of household waste, especially originating from plastic- or plastic-coated materials. The authors collected PM samples emitted from the combustion of variety of plastic materials combusted in a laboratory setup and analysed marker and non-marker compounds using GC/MS. Notable is that the authors show potentially useful plastic combustion markers from both laboratory and ambient PM samples such as the trimer of styrene for polystyrene combustion, melamine for low density fibreboard combustion, and 2-(Benzoyloxy)ethyl vinyl terephthalate for PET combustion.*

*The manuscript is well written and easy to follow. I ask the authors to address one minor comment prior to the publication of this manuscript.*

*Minor comments:*

*I recommend the authors adding the structures of all marker compounds identified in this study (135-TPB, o-TPH, m-TPH, p-TPH, m,p-QTPH, p-QTPH, 2-BEVT, 124-TPB, SSS, ASA, ASS, SSA, SAS, and melamine). This helps readers to relate the marker compounds to a type of plastics combusted in this study.*

The structures of the compounds (except melamine) have been added to the figures. To describe the structure of melamine the following sentence has been modified as follows:

Melamine ($C_3H_6N_6$, heterocyclic aromatic compound) is the component of the melamine-formaldehyde resin…

**Response to Interactive comment of Anonymous Referee #2**

*Comments and questions of the reviewer are in italics*

Authors' answers are in regular typeface

Parts of the answers highlighted in yellow are inserted into the revised manuscript.

The authors thank the referee for the detailed review and comments. The responses are given below.

*In combustion samples, the authors identify organic molecules that could serve as tracers for airborne particulate matter emitted from burning plastic. This article presents data on emissions organic compounds from plastic burning that may serve as chemical tracers of this source in the atmosphere. The article fits within the scope of ACP. The presentation of the article is organized and mostly clear. The scientific methods are described, but are missing validation data in their current form. The stove utilized in these tests should be further described, particularly to provide context to the emissions data and what they represent. The criteria used for recommending that a compound as a tracer needs clarification. Also, the ambient measurements of plastic burning tracers in Bucharest are currently only qualitative, indicating the presence of plastic burning tracers. Prior to publication, these data should be further analyzed quantitatively to gain insight to the amount of garbage burning and the types of plastic that were combusted. Then, more substantial conclusions could be reached.*

1. *In their introduction (line 31), the authors should consider the estimates of waste burning presented by Wiedinmyer et al. in their 2014 article. This modeling study provides the estimates of municipal waste burned around the world (Wiedinmyer et al. 2014).*

Indeed, the modelling study by Wiedenmyer et al., (2014) estimated the emission factors and amount of municipal waste burned by people outside their homes and at dump sites. Here we consider the burning of municipal waste in household stoves which is possibly more difficult to assess since it is a 'hidden' activity. The following sentence has been added to the introduction.

The amount of domestic waste burned in open fires was first estimated in a modelling study by Wiedenmyer et al. (2014) who found annual global waste production to be 2400 Tg, about 26% and 15% of which being burned openly in households and at dump sites, respectively.

2. *In many parts in the manuscript, the authors refer to "illegal waste burning." In many parts of the world, waste burning is not illegal. I encourage the authors to either rethink labeling waste as illegal or to specify the domain that they are discussing.*

The adjective "illegal" has been deleted in all sentences referring to waste burning activities in global terms, but kept in those specifically referring to waste burning in Europe where waste burning is illegal in all countries.

3. *In the introduction at line 89-90 in discussing emission factors of 135-TPB, the authors should include results from Jayarathne et al. (2018). Relevant to this discussion, these authors determined TPB emission factors (among others) from plastic and waste burning (Jayarathne et al. 2018).*

The reference Jayarathne et al., (2018) has been added as suggested, and the sentence has been modified as follows:

Among them 135-TPB is used most commonly as a specific tracer for (mixed) waste burning, and even its emission factor was determined for the open burning of mixed waste (Simoneit et al., 2005), and more recently for the open burning of mixed waste, foil wrappers and plastics (Jayarathne et al., 2018) as well as for the co-burning of PET and PE with firewood in a residential boiler (Tomsej et al., 2018).

4. *The introduction of the paper should describe the stove utilized and its advantages and limitations in this study. In particular, please clarify what the emissions under investigation represent (e.g., geographic region, materials combusted, stove design, etc.) Similarly, please discuss how representative these tests are of real world plastic and combustion.*

The following sentences have been added to the experimental section:

The stove used was a Servant S114 cast iron stove with a heating capacity of 5 kW. It is a commercially available, low cost heating appliance, with a simple stove design, thus it is readily available for the general public for heating their homes with firewood. The simple construction design of the stove also allows the users to combust different items of solid wastes in it. This type of stove is widely used in Central-Eastern Europe, and possibly elsewhere in Europe. Thus our experiments may be considered representative for the burning of small amounts of household waste in low capacity stoves used for domestic heating in Europe.

5. *The term "relative emission factors" is uncommon and somewhat confusing. What the authors present is a mass fraction of organic compounds in PM. "Mass fraction (ug/g)" would be a more clear description.*

As suggested, the term mass fraction ($\mu g\ g^{-1}\ PM^{-1}$) has been added into the manuscript.

6. *The methods description does not yet sufficiently validate the GCMS method. Specifically, the authors should provide an assessment of the accuracy of the method. This is often done by spike recovery samples, in which a known amount of analytes is spiked onto the filter, extracted, and analyzed. The recovered concentration is then reported relative to the spiked concentration as a percent. This quality control test provides a way of determining how much of the target analyte can be recovered in sample preparation.*

Although the differences in the burning conditions may introduce larger uncertainties into the determination of emission factors of a tracer than the analytical methods applied, the latter were estimated by the use of recovery standards. Before the extraction all analysed samples were spiked with a known amount of recovery standard (deuterated terphenyl) to follow the extraction procedure. The average recovery was found to be 93% (RSD=8%). For the tracer compounds measured after derivatization procedures, sedoheptulose was also used to verify the completion of the derivatization reaction.

7. *The utilized internal standard was terphenyl, which has three aromatic rings. It appears that it may be semi-volatile and thus subject to evaporative losses during the extraction (especially drying under nitrogen). This could inadvertently bias the measurements of analytes with lower*

*volatility to be higher. To show that there were not volatile losses, the authors should compare the response of the internal standard recovered from the extraction to that of the internal standard solution as prepared.*

Deuterated terphenyl was used as a recovery standard added to the samples before the extraction, the internal standard was deuterated chrysene added to the samples before the injection. Since the average recovery in the samples was 93% (RSD 8%), it implies that the evaporative losses do not play an important role in the applied sample pre-treatment method.

8.  *The authors recommend many compounds as tracers of plastic burning (e.g., terphenyls at line 195; SSS at 223; section 3.1.5, 3.2). What, specifically are the criteria the authors use to recommend a compound as a tracer? It seems as though the authors seem only to consider the specificity in the samples. Whereas, prior studies have recommended the use of multiple criteria including specificity, gas-to-particle partitioning, atmospheric stability, and detectability in ambient samples.*

Although indeed the other criteria for the potential tracer compounds have not been investigated, the fact that all the suggested compounds were also found in ambient particulate samples implies that they may be worth being considered as potential atmospheric tracers, at least indicatively. As suggested by the reviewer, the most important criteria in our work were the specificity and the confirmed presence of the compounds in ambient particulate samples. We fully agree with the reviewer that further studies on their stability and gas-to-particle partitioning are required to verify their applicability as waste burning tracers.

9.  *The authors should thermodynamically model the gas-to-particle partitioning of recommended tracers. Knowing the fraction in the particle phase at ambient temperature and pressure would build support for their use as PM10 tracers.*

The gas-to-particle partitioning was not modelled, but the equilibrium vapour pressures of the compounds were estimated using US EPA (2012). The estimated vapour pressures of the identified compounds are in the range between $3.87 \times 10^{-11}$ mmHg at 25°C and $1.64 \times 10^{-5}$ mmHg at 25°C, the latter is lower than that for e.g. the syringic aldehyde ($6.49 \times 10^{-5}$ mmHg at 25°C) suggested as a biomass burning tracer by Simoneit et al., (2002) and is comparable with that of terephthalic acid ($1.19 \times 10^{-5}$ mmHg at 25°C). We also note here that the estimated vapour pressure of SSS and 2BEVT ($3.34 \times 10^{-7}$ mmHg at 25°C and $7.51 \times 10^{-7}$ mmHg at 25°C, respectively) is very close to that of levoglucosan ($3.47 \times 10^{-7}$ mmHg at 25°C), the most widely used biomass burning tracer compound.

US EPA (2012). Estimation Programs Interface Suite™ for Microsoft® Windows, v 4.11. United States Environmental Protection Agency, Washington, DC, USA.

Simoneit, B. R. T.: Biomass burning - A review of organic tracers for smoke from incomplete combustion, Appl. GEOCHEMISTRY, 17(3), 129–162, doi:10.1016/S0883-2927(01)00061-0, 2002.

10. *The authors should quantitatively assess the analytical uncertainties associated with the measurements of these compounds in source and ambient PM10 samples.*

==Based on the recoveries and repeated measurements the estimated uncertainty of the analytical measurements is about 20%.== Here we note that the overall uncertainties of the results (the mass fraction of the tracer compounds in the emitted PM10 and the ambient concentrations of the tracers) are considerably higher due to the unavailability of certain standards (thus the calibration can only be based on structurally similar compounds), as well as the large effect of the burning conditions on the pyrolysis process that produces the tracer compounds.

11. *In the Bucharest samples (section 3.3), were these tracers identified based on their retention index or retention time of a commercial standard?*

In the Bucharest samples the tracers were identified based on their retention time and the presence and ratio of the target and qualifier ions. Since commercial standards were not available for most of the compounds, these parameters were determined from the samples obtained from burning of different plastics in the laboratory. In these emission samples obtained from the burning experiments the presence of the compounds were confirmed based on their retention indices and mass spectra.

12. *Neither the methods nor the results sections report the date, year or season of the Bucharest PM10 sample collection. Please add this information, as appropriate, to each section. Also, please add discussion of any seasonal and meteorological significance (e.g., wintertime, stagnant conditions, etc.).*

Information on the Bucharest samples (collection day and meteorological conditions) is now included in the experimental section.

The section 2.1 was completed as follows:
Ambient samples were collected ==in winter on 14, 17 and 25 February 2020== in Magurele (Atmosferei Street), Romania, located about 12 km far from Bucharest centre. The PM10 samples were collected on quartz filters (Advantec QR-100) with Digitel DHA-80 high volume samplers. The sampling time was 24 hours starting from midnight for each sample. ==The meteorological situation was characterized by a high-pressure regime on 14 and 17 February, and with low pressure on 25 February. The temperature values were above 0 ºC during the daytime, and were slightly below 0 ºC during the night.==

13. *What were the PM10 concentrations in these ambient samples? The tracer-to-PM10 ratio is useful to compare to the mass fractions in the source emissions.*

The PM10 concentrations of the ambient samples are added to the table. The authors agree with the reviewer that it is useful to compare the tracer to PM10 ratios obtained for the ambient samples with the ones of the emission sources, but in this paper our objective was to propose new specific tracer compounds for solid waste burning in residential stoves, and to verify their presence in ambient samples. Our objective was not to estimate the contribution of waste burning to the ambient PM10 concentrations, as this will be the subject of a follow-up paper involving a large number of particulate samples collected at several locations in Hungary and Romania.

*14. To Table 1, please add the target ion/qualifier ion ratio observed for commercial standards (i.e. in calibration solutions). This will give an indication of the value in a sample with minimal sample matrix.*

The target ion/qualifier ion ratios of the available standards have been added to Table 1.

*15. At line 324-324 a sentence begins "It can be seen…." And asks the reader to interpret Table 1. To help the reader in doing so, please mark the compounds that have good agreement versus those that do not, so that this is clear.*

Table 1 has been modified as requested, the ion intensity ratios which differ considerably (the difference is larger than 22%) from that obtained for the source samples are now underlined and in italics.

The sentence has been also modified as follows:

It can be seen that the intensity ratios of the target ion and the qualifier ion of a few tracer compounds (marked underlined and in italics in Table 1 for the m-terphenyl (m-TPH), SSA, and for the ASA in one sample)…

*16. It is suggested that the intensity ratios of the target and qualifier ions across the ambient samples and emission samples vary because of isobaric interferences coming from the sample matrix. Can the authors quantitatively assess how this bias may impact the reported concentrations of the analytes in Tables 1 and S2?*

In the text, we have pointed out that the fragmentation is smaller for the aromatic compounds, thus the lower intensity ions and of course their ratio may be affected by the background (and/or co-eluting compounds). For quantification, the signal of the target ion is integrated, which is normally the molecular ion of the aromatic component therefore its intensity is the highest, thus the effect of the other compounds is the lowest possible.

Table 1 contains the ion intensity ratios of the available standards as well. The good agreement of the intensity ratios between the available standards and the components measured in the source samples indicates that the effect of the background on the concentration is low.

*17. At present, the ambient measurements are very qualitative and are discussed only in terms of their presence in 3 particular samples. These data should be analyzed more quantitatively to gain insight to the amount of garbage burning and the types of plastic that were combusted.*

With the qualitative analysis of the 3 samples from Bucharest, our primary aim was to demonstrate that the proposed tracer compounds are indeed present not only in the source samples, but also in the ambient samples. As indicated above, the assessment of the potential contribution of waste

burning to PM10 pollution at different locations will be the subject of a follow-up paper based on a much larger number of samples, locations, and a comprehensive statistical analysis.

18. *In section 3.3, can the ambient measurements be used to estimate the contribution of garbage burning to PM10?*

In this manuscript our objective was to propose new specific tracer compounds for solid waste burning in residential stoves, and to verify their presence in ambient samples. Our objective was not to estimate the contribution of waste burning to the ambient PM10 concentrations, as this will be the subject of a follow-up paper involving a large number of particulate samples collected at several locations in Hungary and Romania.

19. *Also in section 3.3, to what extent do the ratios of the plastic burning tracers indicate the types of plastic that were likely burned? In section 3.2, the relative amounts of established tracers were shown to vary with plastic type. What types of plastic are suggested by the relative amounts of these tracers in ambient PM10?*

Since the proposed aromatic compounds used to calculate the different ratios are not specific tracers for the burning of a given waste type, thus the application of their relative ratios in ambient particulate samples for emission source type assessment require highly complex statistical calculations in combination with the use of highly specific tracers on a much more extensive dataset, which is outside the scope of this manuscript.

20. *To Table S2, please add PM10 emission factors, as well as organic and elemental carbon (OC and EC). These data would help in comparing these measurements to ambient and source samples in future studies.*

The PM10 emission factors were published in our previous paper (Hoffer et al., 2020), the OC and EC were not measured in the samples.

Hoffer, A., Jancsek-Turóczi, B., Tóth, Á., Kiss, G., Naghiu, A., Levei, E. A., Marmureanu, L., Machon, A., and Gelencsér, A.: Emission factors for PM10 and polycyclic aromatic hydrocarbons (PAHs) from illegal burning of different types of municipal waste in households, Atmos. Chem. Phys., 20, 16135–16144, https://doi.org/10.5194/acp-20-16135-2020, 2020

21. *Figures – I agree with the other referee that seeing chemical structures for these organic molecules would be helpful. (I had to look them all up when reading, because they are new to our field and are not yet commonly known.) One way to incorporate these into the paper is to include them in the existing figures in which their concentrations are shown. There appears to be ample white space for these to be added.*

As requested, the chemical structures of the compounds are now added to the figures.

22. *Figures – The grayscale color scheme used in the figures is difficult to follow. In particular, the first and third gray colors are practically indistinguishable. Can the 1st be changed to black and the 3rd be changed to white, perhaps? This would provide better contrast.*

The colour scheme of the figures has been changed; the colour of the last column of the figures containing at least 3 datasets has been changed to white.

23. *For many of the figures, only the sample(s) with the highest concentration(s) can be seen. To show the features of the samples with low concentrations, could these data be shown on a logarithmic scale?*

The linear representation was used to highlight the types of wastes that emit large amounts of the given component during the burning. Plots with logarithmic scales would not help to get a more accurate reading. The data are summarised and can be viewed in Table S2.

24. *Please check, is it "rag" or "RAG"?*

The word "rag" has been changed to RAG.

25. *Copy editing needed, some grammatical errors, other times words are missing.*

Grammatical errors and missing words were corrected in the manuscript.

26. *Figure captions – superscripting missing for units in Figure 1.*

Superscripts of the units in the figure captions have been corrected.

---

## Author Response (AR2)

Dear Editor,

all the changes your have requested has been completed and the changes are  hallmarked in turquoise colour in the revised manuscript uploaded.

Best regards

András Gelencsér